# Impact of 5-HT_6_ Receptor Subcellular Localization on Its Signaling and Its Pathophysiological Roles

**DOI:** 10.3390/cells12030426

**Published:** 2023-01-27

**Authors:** Séverine Chaumont-Dubel, Sonya Galant, Matthieu Prieur, Tristan Bouschet, Joël Bockaert, Philippe Marin

**Affiliations:** Institut de Génomique Fonctionnelle, Université de Montpellier, CNRS, INSERM, 34094 Montpellier, France

**Keywords:** G-protein-coupled receptor, serotonin, signaling, primary cilium

## Abstract

The serotonin (5-HT)_6_ receptor still raises particular interest given its unique spatio-temporal pattern of expression among the serotonin receptor subtypes. It is the only serotonin receptor specifically expressed in the central nervous system, where it is detected very early in embryonic life and modulates key neurodevelopmental processes, from neuronal migration to brain circuit refinement. Its predominant localization in the primary cilium of neurons and astrocytes is also unique among the serotonin receptor subtypes. Consistent with the high expression levels of the 5-HT_6_ receptor in brain regions involved in the control of cognitive processes, it is now well-established that the pharmacological inhibition of the receptor induces pro-cognitive effects in several paradigms of cognitive impairment in rodents, including models of neurodevelopmental psychiatric disorders and neurodegenerative diseases. The 5-HT_6_ receptor can engage several signaling pathways in addition to the canonical Gs signaling, but there is still uncertainty surrounding the signaling pathways that underly its modulation of cognition, as well as how the receptor’s coupling is dependent on its cellular compartmentation. Here, we describe recent findings showing how the proper subcellular localization of the receptor is achieved, how this peculiar localization determines signaling pathways engaged by the receptor, and their pathophysiological influence.

## 1. Introduction

G-protein-coupled receptors (GPCRs) are highly versatile proteins that can be activated by a plethora of stimuli (photons, lipids, amino acids, amines, peptides, and proteins) and can control a large number of physiological processes. They are therefore therapeutic targets of choice for a wide variety of pathological conditions. As indicated by their name, these receptors signal through the activation of G-proteins. In addition to canonical G-protein-dependent signaling, GPCRs can also engage various downstream effectors in a G-protein-independent manner [1,2]. These diverse pathways can be selectively activated by biased ligands [3]. They also depend on receptor association with a large diversity of intracellular proteins (designated as GPCR-interacting proteins or GIPs), as well as on the subcellular compartmentation of the receptor [1,4]. Several GPCRs also display constitutive activity in addition to agonist-dependent activation [5,6]. Constitutive activity can be of great importance for a GPCR function [7], and thus represents an area of extensive research, including for orphan receptors [8].

Among the 14 serotonin (5-HT) receptor subtypes, the 5-HT_6_R is the only one whose expression is restricted to the central nervous system. Its subcellular localization is also unique for a serotonin receptor since it is predominantly detected in the primary cilium [9,10,11,12,13]. The efficiency of 5-HT_6_R antagonists to alleviate cognitive impairments in a number of rodent models of neurodevelopmental, psychiatric, and neurodegenerative diseases [14,15,16,17,18,19,20,21,22,23,24,25,26,27,28,29,30,31], in addition to its absence in peripheral organs, makes it a therapeutic target of choice for the treatment of cognitive symptoms of schizophrenia, autism spectrum disorders, and dementia associated with Alzheimer’s disease. These findings, together with the role of the receptor in neuropathic pain [32], epilepsy [33,34], and the regulation of food intake [35], prompted a search for new specific ligands exhibiting antagonist (or inverse agonist) properties [36]. The receptor is also a major player in several key neurodevelopmental processes, such as neuronal migration and morphogenesis of dendritic tree, suggesting that the disruption of its normal function could take part in the pathophysiology of neurodevelopmental diseases. Indeed, Duhr et al. demonstrated that the 5-HT_6_R interacts with the activated cyclin-dependent kinase Cdk5. Cdk5 can phosphorylate the C-terminal part of the receptor on the serine 350. This phosphorylation of the receptor by Cdk5 is also necessary for pyramidal cortical neurons to undergo the multipolar to bipolar transition which precedes radial migration. It then regulates the speed of the radial migration [37]. This phosphorylation also results in the activation of Cdc42, leading to the induction of neurite growth [38]. The receptor is then dephosphorylated, which allows for its interaction with the G-protein-regulated inducer of neurite growth 1 (GPRIN1). This interaction is necessary for the elongation and complexification of the dendritic tree [39].

The 5-HT_6_R is canonically coupled to the protein Gs [40], but also signals through Gq [41] and activates Fyn [42], Jab1/Jun kinase [43], mTOR [15], and Cdk5-Cdc42 pathways [38]. The 5-HT_6_R also displays a high level of constitutive activity towards the Gs pathway in vitro and in vivo [44,45], but also towards the Cdk5 pathway [38], and the mTOR pathway [46].

Here, we will review the latest data on the subcellular localization of the 5-HT_6_R and highlight how its peculiar subcellular localization determines the nature of receptor-operated signaling. We will pay particular attention to the processes controlled by ciliary receptors and to receptors expressed outside of the cilium.


**(1). Predominant ciliary localization of the 5-HT_6_R in neurons and astrocytes**


Since its discovery in the early 1990s, being able to precisely locate the 5-HT_6_ receptor (5-HT_6_R) has proven challenging, since no specific antibody enabling the detection of the endogenous protein is available. qPCR and in situ hybridization were first used to describe the overall pattern of expression of the receptor. These techniques allowed the mapping of areas of the brain where the receptor is expressed, but they did not reveal the subcellular localization of the protein. At the end of the 1990s, an antibody targeting the C-terminal part of the receptor was raised in rabbits and used to perform immunolabeling in rat brains. These studies described its presence in the dendritic compartment, but also in the primary cilium [9,10,47]. However, the specificity of the antibody used in those studies was a concern, since it revealed a strong expression of the receptor in areas where no mRNA was found, such as in the cerebellum [48]. Nevertheless, the predominant ciliary localization of the receptor in embryonic and adult brains was further confirmed by several studies in vitro and in vivo [11,49,50]. The latter study also revealed the presence of the receptor in the primary cilium of both neurons and astrocytes [11].

The primary cilium is a non-motile organelle, which plays essential roles in neurodevelopment, from the regulation of migration to cell fate determination [51,52,53,54]. In addition to its neurodevelopmental roles, the primary cilium from both neurons and astrocytes is crucial for adult brain function [55,56]. The disruption of neuronal cilia has major pathological consequences. In humans, ciliopathies are genetic disorders that originate from mutations in genes coding for proteins that participate in the formation of the primary cilium or of the basal body, i.e., its anchoring platform. These mutations alter the primary cilium function. Ciliopathies are characterized by neuro-developmental abnormalities, as well as by cognitive and learning deficits [57,58]. Many GPCRs as well as GPCR effectors, such as adenylate cyclase and ß-arrestins, can be found in the primary cilium where GPCR signaling is still extensively studied [12,59,60].

Pluripotent embryonic stem cells possess a primary cilium, but it does not appear to be stable. Recently, Phua et al. established a link between primary cilium decapitation and the cell cycle [61]. ESCs undergo a very rapid cell cycle. It is likely that the primary cilium is quickly decapitated in these cells, making it hard to observe in a culture. Upon differentiation and cell cycle exit, the frequency of decapitation will be reduced, and cells will exhibit a stable primary cilium [61,62]. Interestingly, in a transgenic embryonic stem cell line engineered to express the 5-HT_6_R-GFP under the control of doxycycline, the 5-HT_6_R is primarily localized at the membrane of the cell body, but as soon as the cells are differentiated, the receptor is addressed to the newly formed primary cilium [11]. This change in the receptor’s subcellular localization has been demonstrated in the two-dimension culture of these cells, but these cells can also be cultivated as brain organoïds in three dimensions (Figure 1A). Under these conditions, the induction of the 5-HT_6_R expression also results in its trafficking to the primary cilium, as shown by its colocalization with the ciliary marker Arl13b (Figure 1B). In these cultures, the primary cilia expressions of the 5-HT_6_R are significantly longer than those found in cells where the receptor has not been induced (Figure 1B). This increase in cilia length strongly suggests a role for the receptor based on the morphology and function of the primary cilium, which remains to be elucidated.


**(2). Identification of 5-HT_6_R cilium-targeting domains**


The 5-HT_6_R is the only serotonin receptor displaying a ciliary location, suggesting a special role for this receptor in neurodevelopment and brain function. The localization of GPCRs in the primary cilium is a highly controlled phenomenon. Different types of ciliary-targeting sequences (CTSs) have been identified. A VxPx motif was found on the C-terminal tail of the rhodopsin [63]. A [R/K][I/L]W motif was identified on the third intracellular loop (ic3) of the NPY2R and GPR88 [64]. Berbari et al. sought to identify the ciliary-targeting sequences of the SSTR3 and 5-HT_6_R. Using a domain transfer approach, the authors showed that the ic3 sequence of both SSTR3 and 5-HT_6_R was sufficient to target the primary cilium SSTR5 and 5-HT_7_R receptors—two non-ciliary GPCRs related to SSTR3 and 5-HT_6_R, respectively [65]. These results suggest that the Ic3 of both receptors contained a CTS. Based on analogy with known CTSs, they hypothesized that the CTS could be a tandem of Ax[S/A]xQ motifs, the ic3 of the somatostatin receptor 3 (SSTR3), and the AxAxQ motif in the ic3 of the 5-HT_6_ R [65]. Later, Barbeito et al. performed a mutagenesis study which revealed that mutating a RKQxxxV motif in the ic3 of the 5-HT_6_R partially disrupted its targeting to the primary cilium [32]. Interestingly, there was an alanine upstream of the arginine, giving rise to an AxxxQ motif, but the importance of this alanine has not yet been investigated [65]. However, mutating the CTS on the Ic3 of SSTR3 or the 5-HT_6_R did not result in a complete loss of ciliary localization, suggesting the presence of additional molecular determinants located out of the ic3 that contributed to their targeting to the cilium. A similar domain-swapping approach led to the discovery of a second CTS sequence in the C-terminal part of these two GPCRs [32]. These observations suggest that the 5-HT_6_R ciliary localization depends on two redundant CTSs—the RKQxxxV motif in the ic3 (CTS1) and an additional LPG motif in a leucine-rich area of the C-terminal tail (CTS2) [32]—with each motif alone being sufficient to target at least part of the receptor to the primary cilium. CTS1 promotes the binding of the 5-HT_6_R to RABL2, a guanine nucleotide exchanger which also regulates the trafficking of GPCRs to the cilium. In contrast, CTS2 regulates the interaction of the 5-HT_6_R with the Tubby-like protein 3 (TULP3). Surprisingly, disrupting the LPG motif strengthens the interaction with TULP3 rather than weakening it. Residues near the LPG motif seem to be involved in the recruitment of TULP3, but not in the targeting of the 5-HT_6_R to the primary cilium. Barbeito et al. postulated that the CTS2 could be required to release TULP3 once the receptor had entered the primary cilium [66], but this has not yet been demonstrated. How TULP3 regulates the trafficking of the 5-HT_6_R to the cilium remains unknown [66]. The effect of mutating either or both CTSs is summarized in Figure 2.


**(3). Identification of axo-ciliary serotonergic synapses**


Using focused ion beam–scanning electron microscopy (FIB-SEM) to reconstruct the microenvironment of neuron primary cilia, Sheu et al. showed the presence of axonal varicosities reminiscent of classical presynaptic axonal terminals in the vicinity of most cilia of CA1 pyramidal neurons [41]. Labelling serotonergic axons with an anti-serotonin transporter antibody revealed that cilia are juxtaposed with serotonergic axons in ~50% of the axo-ciliary structures detected by FIB-SEM. Furthermore, all axonal sites juxtaposed with cilia are synaptophysin-positive, suggesting that they can release serotonin. Consistent with this hypothesis, optogenetics and chemogenetics studies using a cilia-targeted serotonin sensor and channelrhodopsin showed that the activation of serotonergic axons induces a serotonin release onto neuronal cilia, leading to the activation of the ciliary 5-HT_6_R. Collectively, this elegant series of experiments identified axo-ciliary serotonergic synapses expressing postsynaptic ciliary 5-HT_6_Rs that might underlie some of the pathophysiological functions of these receptors.


**(4). The ciliome and its role in 5-HT_6_R signaling in the primary cilium**


Kohli et al. used a biotin proximity-labeling approach based on the 5-HT_6_R to target the APEX2 enzyme to the primary cilium and tandem mass spectrometry to identify the membrane-associated ciliary proteome in an unbiased manner. They also deciphered the specific interactome of ciliary 5-HT_6_Rs via a pull-down approach in cells expressing 5-HT_6_R-APEX-GFP [67]. They identified 49 candidate partners of the receptor and showed that most of the proteins interacting with the receptor in the primary cilium were actin-binding proteins, suggesting a role for the receptor in the regulation of the actin cytoskeleton. This link with the actin cytoskeleton is consistent with several studies indicating that the expression and activity of the receptor control the cilium morphology. For instance, the treatment of cultured striatal neurons with a selective antagonist of the 5-HT_6_R significantly reduced the length of the primary cilium [49]. Moreover, the primary cilium of striatal neurons prepared from Htr6-KO mice was shorter than that of neurons from wild-type mice [49]. This phenotype was rescued by the transfection of the 5-HT_6_R in KO neurons. Likewise, Dupuy et al. demonstrated that the length of primary cilia in brain slices from Htr_6_-KO mice was reduced, compared to WT mice, indicating that the 5-HT_6_R controls cilium morphology in vivo [11]. Notably, the expression of the 5-HT_6_R in primary striatal neurons from KO mice also increased the dendrite length [50], in line with previous findings indicating that the receptor promotes the initiation of neurite growth and neurite extension [38,39].

Going further into the physiopathological roles of the receptor in the primary cilium, Hu et al. studied the effect of the overexpression or knock-down of the 5-HT_6_R in neurons prepared from WT or APP/PS1 mice, a mouse model of Alzheimer’s disease [19]. Corroborating these studies, the overexpression of the receptor induces an elongation of the primary cilium, and sometimes the appearance of extra branches in cultured hippocampal neurons from WT mice. Conversely, knocking down the receptor’s expression using siRNA reduced cilia length. To further explore the molecular determinants by which the receptor regulates the length of the primary cilium, the authors used several mutants which affect the receptor’s function. Mutating the D106 residue of the 5-HT_6_R will result in a receptor that cannot be activated by an agonist, but still displays constitutive activity towards Gs [37]. The receptor bearing mutations in the F69, T70, and D72 sequence will not activate G-protein at all [68], losing both its evoked and constitutive activity. The authors showed that mutating D72; D106; and a combination of F69, T70, and D72 is key for the regulation of cilium morphology, which suggests that evoked activity rather than constitutive activity is responsible for the effect of the receptor on the cilia length. Furthermore, they demonstrated that the primary cilia of hippocampal neurons from APP/PS1 mice were longer than those of WT mice, an observation which correlated with the increased expression of the 5-HT_6_R in APP/PS1mice. Finally, the authors used SB271046, a compound that exhibits antagonist and inverse agonist properties toward the Gs pathway and showed it could restore cilia morphology and cognitive deficits observed in APP/PS1 mice [19]. Whether 5-HT_6_Rs affect cognition in Alzheimer’s disease through the regulation of primary cilia morphology certainly warrants further exploration.

Another link between the 5-HT_6_R and actin cytoskeleton stems from the study of Sheu et al., which shows that treating cultured hippocampal neurons with the 5-HT_6_R antagonist SB742457 triggers the translocation of the actin-binding protein adducin 1 to the nucleus. Similarly, nuclear adducin 1 is denser in Htr6-KO mice compared to WT mice. This suggests that ciliary 5-HT_6_R signaling might affect nuclear actin and, consequently, the global chromatin state. Using an ATAC-see approach to label accessible chromatin, the same study demonstrated that the activation of the 5-HT_6_R results in an increased ATAC (see labeling), indicative of a more open chromatin state. Conversely, comparing the chromatin state in WT to the chromatin state in Htr6-KO mice revealed a more compact, less active chromatin in the KO mutants. Finally, the activation of ciliary 5-HT_6_Rs resulted in an increase in H4K5 acetylation but not in H3K27 acetylation [41]. Collectively, these results suggest that 5-HT_6_Rs located in the primary cilium are linked to both cytoplasmic actin cytoskeleton and nuclear actin to control the morphology of the primary cilium, as well as the chromatin state and specific epigenetic mechanisms.


**(5). Expression of the receptor outside the cilium**


Overexpression of the 5-HT_6_R in neurons or in mouse inner medullary collecting duct (mIMCD3) cells can lead to ectopic receptor localization in the somato-dendritic compartment [44,50]. While the endogenous 5-HT_6_R is mainly detected in the primary cilium in embryonic and adult neurons, the high concentration of 5-HT_6_Rs in this organelle results in a very strong immunofluorescence masking signal from compartments expressing the receptor at lower levels. Using a tyramide booster, which amplifies the fluorescence signal from the receptor in a very specific way, we showed a slight expression of the 5-HT_6_R in other neuronal compartments, including the cell body and dendritic spines [28].

Using 5-HT_6_R-GFP knock-in mice, we found that, in the embryo, the receptor is located in the primary cilium [11]. Intriguingly, during early post-natal life (post-natal day 1 to 9), the 5-HT_6_R is mainly found in the cell body of neurons in certain regions of the brain, including the striatum and deep cortical layers. The receptor then returns to the primary cilium at around post-natal day 10, where it remains for the rest of the animal’s life [11]. The post-natal period corresponds to the time when neurons in these areas initiate neurite growth and branching. We previously demonstrated that these two phenomena require the sequential interaction of the receptor with the cyclin-dependent kinase 5 (Cdk5) [38] and the G-protein-regulated inducer of neurite growth 1 (GPRIN1) [39], i.e., two proteins located outside the cilium. We thus speculate that this transient relocation of the receptor to the somatodendritic compartment at the neonatal stage might be a key mechanism underlying its modulation of dendritic tree morphogenesis and the establishment and refinement of neuronal connectivity. The transient receptor relocation to the membrane of the cell body during the neonatal days was not due to receptor overexpression at that developmental stage, as qPCR experiments showed no correlation between its expression level and its neuronal compartmentation during brain development. Furthermore, contrasting with the observations made in neurons overexpressing the 5-HT_6_R [44,50], in the striatum and hippocampus of 5-HT_6_R-GFP-KI mice, we showed that cells expressing the receptor in the soma seldom presented a labeled primary cilium. Co-labeling with AC3 and Arl13b—two markers of primary cilia—confirmed that most neurons lack a mature primary cilium between P1 and P9. This absence of primary cilium is consistent with studies showing that ciliogenesis in pyramidal neurons starts around birth and continues during the first few weeks of life [52]. To confirm the hypothesis that the receptor’s somato-dendritic localization in post-natal neurons results from the lack of primary cilium, we generated embryonic stem cells expressing an HA-5-HT_6_R-GFP construct under a doxycycline-inducible promoter. We demonstrated that in undifferentiated stem cells, very few primary cilia are already formed and, accordingly, that the 5-HT_6_R is mainly located in the plasma membrane of the cell body. However, when the 5-HT_6_R expression was induced in differentiated cells that possess a primary cilium, the receptor was almost exclusively localized in the cilium. This differential subcellular localization is reminiscent of what was observed in vivo and suggests that the receptor will preferentially locate to the primary cilium when it is formed, whereas it relocates to the membrane of the cell body in immature neurons lacking a primary cilium. Consistently, in the superficial layers of the cortex that harbor the earliest-born neurons, we observed that almost all neurons are already ciliated at post-natal day 1, and that the receptor exhibits a ciliary localization in these areas.


**(6). Cell-compartment-specific 5-HT_6_R-operated signaling**


GPCRs can be coupled to different G-proteins, displaying biased signaling, depending on their subcellular localization or on the ligand that activates them. In the case of the 5-HT_6_ receptor, the canonical signaling pathway described in the early 1990s is a coupling to Gs [40]. This coupling can be found when the receptor is expressed in heterologous systems [44,69], but also primary striatal neurons [45]. Using a biosensor to directly visualize cAMP production in response to Gs activation, Jiang et al. studied the coupling of the 5-HT_6_R to Gs in mIMCD3 cells when it was both strictly and loosely targeted to the primary cilium [44]. After an overnight treatment with the smoothened agonist (SAG) to activate the Hedgehog (Hh) pathway, both ciliary and somatic 5-HT_6_R activation resulted in cAMP production. In contrast, when the Hh pathway was not activated, 5-HT_6_R-dependent cAMP production was detected only in the cell body, not in the primary cilium. This specificity in the receptor’s coupling was observed for both the agonist-evoked receptor activation of Gs and agonist-independent (constitutive) receptor activity, indicating that, in these conditions, coupling to Gs is restricted to receptors on the cell body [44].

Using a cilia-targeted FRET-based RhoA biosensor, Sheu et al. demonstrated that the 5-HT_6_R expressed in the primary cilium activates the Gq/11-Trio-RhoA pathway [41], consistent with an interactomics screen which indicates that the receptor can recruit Gq/11 proteins [38]. Activation of the Gq/11-Trio-RhoA pathway results in the phosphorylation of adducin-1 by the Rho-associated kinase and increases its affinity for F-actin, an effect that may affect its nuclear translocation and nuclear actin, which will in turn alter the chromatin structure and epigenetic marks (see paragraph 4).

The 5-HT_6_R is also coupled to Cdk5. The interaction between Cdk5 and the 5-HT_6_R results in the receptor’s phosphorylation, and in the initiation of neurite outgrowth through the activation of the small GTPase cdc42 [38]. The elongation and branching of neurites require the dissociation of Cdk5 from the receptor and its replacement by GPRIN1. This new interaction potentiates the receptor’s constitutive activity towards Gs and activates the PKA pathway [39]. Both Cdk5 and GPRIN1 are located outside the primary cilium, and we showed that the interaction between the receptor and GPRIN1 occurs at the membrane of the cell body and not in the primary cilium in the neonatal brain [39]. As previously mentioned, the 5-HT_6_R transiently relocates from the primary cilium to the cell body during the first post-natal week. This transient relocation exemplifies how the dynamic regulation of 5-HT_6_R compartmentation in neurons during brain development allows it to have different physiological functions, by switching partners and signaling pathways. The different signaling pathways engaged by the receptor, depending on its subcellular localization or its developmental stage, are described in Figure 3.

## 2. Conclusions

Unlike other serotonin receptors found throughout the body, the 5-HT_6_R expression is restricted to the central nervous system where it shows the highest densities in brain regions involved in higher-level cognitive functions, such as the striatum, hippocampus, and prefrontal cortex, making it an attractive target for the treatment of cognitive deficits associated with several psychiatric disorders of neurodevelopmental origin, as well as of neurodegenerative diseases. At the subcellular level, the 5-HT_6_R is the only serotonin receptor to be addressed to the primary cilium of neurons in embryonic life and adulthood, while it is transiently located at the membrane of the cell body during the first postnatal week. Depending on time and its subcellular location, this receptor can engage different signaling pathways. The physiological outcomes are varied. During development, it has a key influence on neuronal migration and dendritic tree morphogenesis. In adults, it can regulate gene expression by controlling the chromatin state through the epigenetic modification of histones. The dynamic spatio-temporal regulation of receptor signaling, depending on its subcellular compartmentation and association with different protein partners, likely explains the large diversity of processes under the control of this unique and versatile receptor from early developmental to the adult stage.

## Figures and Tables

**Figure 1 cells-12-00426-f001:**
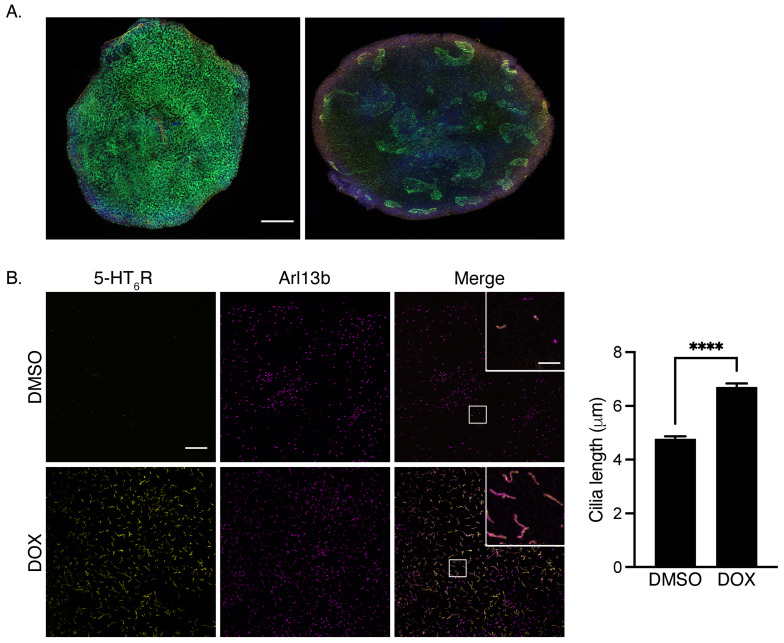
(**A**) Confocal images (20×) of 3D cultures (brain organoids) generated from ES cells expressing either cytosolic GFP (left) or 5-HT_6_R-GFP (right). The GFP or 5-HT_6_R are shown in green, nuclei are labeled with DAPI, and primary cilia are labeled with Arl13b (red). Scale bar 200 μm. (**B**) Confocal images (63×) of cells treated with either DMSO (no induction of the 5-HT_6_R expression) or with doxycycline (DOX, induction of the 5-HT_6_R expression) for 24 h, 7 days after starting differentiation. The 5-HT_6_R (yellow) is expressed in primary cilia, as shown by colocalization with the ciliary marker Arl13b (magenta). The inset on the merge image shows a 10× zoom on cilia labeled with Arl13b and GFP (Scale bar 20 μm). The histogram on the right shows that cilia length is significantly increased in cells expressing the 5-HT_6_R (DOX) when compared to DMSO-treated cells (6.70 ± 0.13 μm (n = 558) for DOX-treated cells vs. 4.76 ± 0.09 μm (n = 460) for DMSO-treated cells). **** *p* < 0.0001, unpaired *t*-test. Scale bar 20 μm.

**Figure 2 cells-12-00426-f002:**
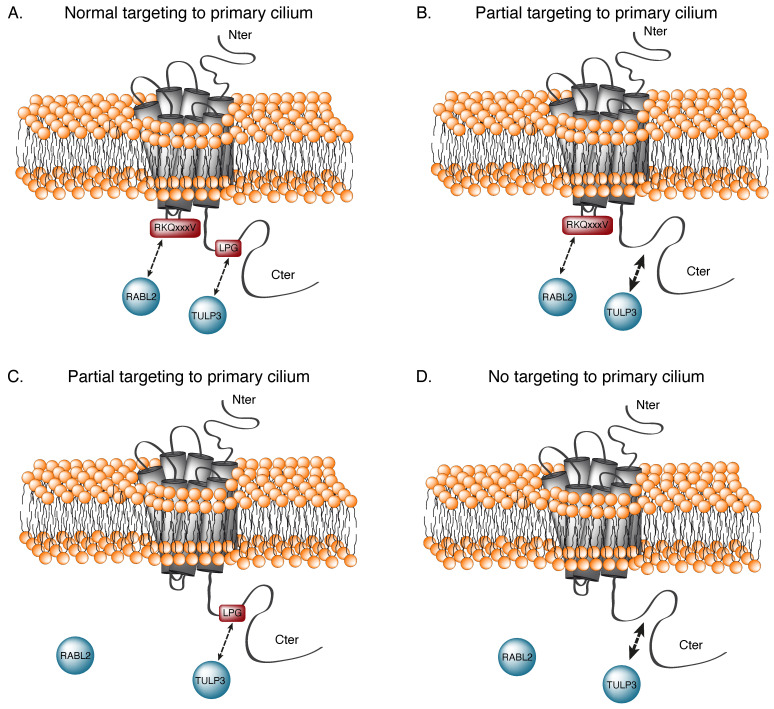
The 5-HT_6_R possesses 2 cilia-targeting sequences (CTSs). When present, CTS1 (on the ic3 loop) interacts with the RABL2 GTPase, whereas CTS2 (on the C-terminal domain) interacts with TULP3. Both CTSs are required for normal cilia targeting (**A**). Cilia targeting is only partial when CTS2 is mutated, but paradoxically, interaction with TULP3 is increased rather than lost (**B**). When CTS1 is mutated, interaction with RABL2 is lost, and cilia targeting is diminished (**C**). Finally, mutating both CTSs results in a complete loss of ciliary localization of the receptor (**D**).

**Figure 3 cells-12-00426-f003:**
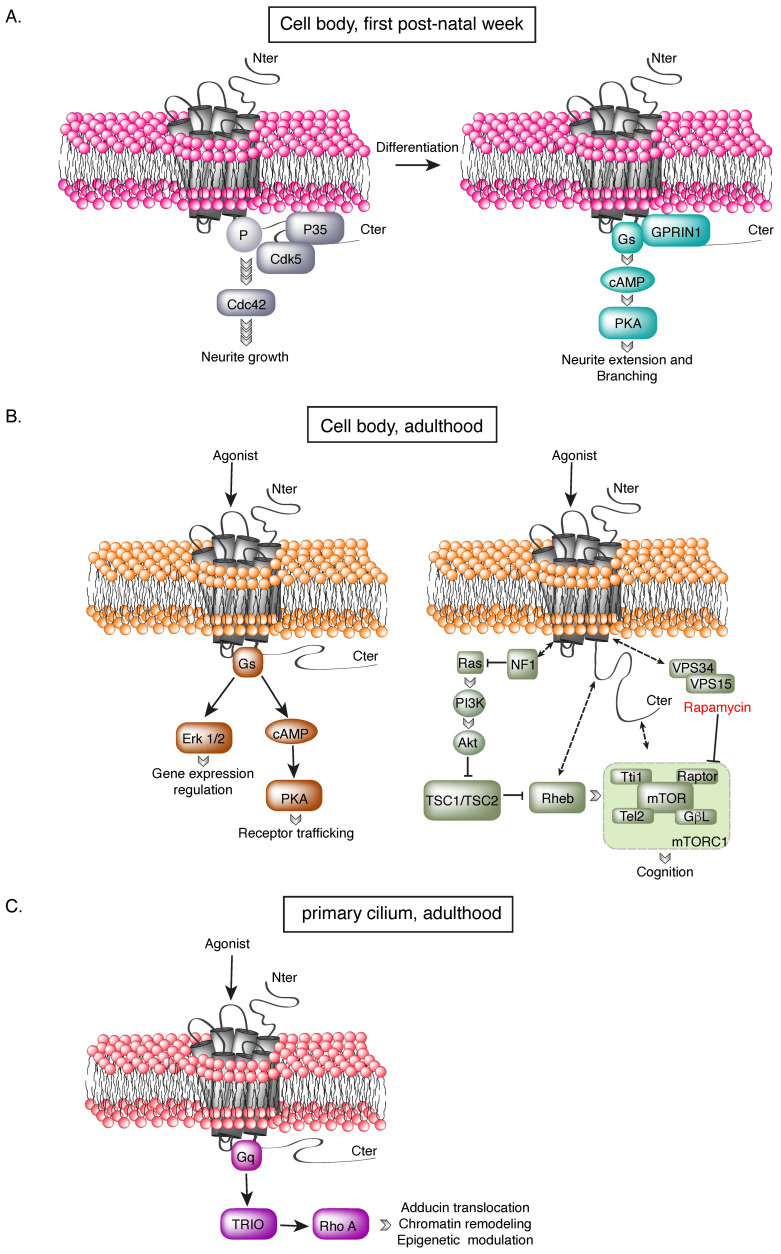
Spatio-temporal characteristics of the 5-HT_6_R signal transduction. Depending on its subcellular localization or developmental stage, the 5-HT_6_R can be coupled to different signaling pathways. (**A**) During the first post-natal week, the receptor is expressed at the membrane of the cell body. This relocation of the receptor allows for its coupling to Cdk5, which will phosphorylate the receptor and induce differentiation and neurite outgrowth through a Cdc42-dependent pathway. The receptor will subsequently couple to GPRIN1, to promote neurite elongation and branching through a Gs-cAMP-PKA-dependent signaling pathway. The interaction with Cdk5 and GPRIN1 is sequential; GPRIN1 cannot interact with the receptor phosphorylated by Cdk5 [39]. (**B**) At the adult stage, the somatic 5-HT_6_R can couple to Gs. Its association with different GIPs (G-protein-coupled receptor-interacting proteins), such as Fyn or JAB1, will modulate the receptor plasma membrane localization and its regulation of gene expression through the activation of the MAP kinase pathway. The receptor can also activate the mTOR pathway, which underlies its roles in cognition. Whether both signaling pathways can couple to receptors in the same conformational state remains to be determined. (**C**) The 5-HT_6_R is primarily located in the primary cilium, where it couples to the Gq-TRIO-RhoA pathway. Upon activation by serotonin from a nearby axon, the receptor in axo-ciliary synapse inhibits adducin-1 nuclear translocation, induces chromatin opening, and modulates epigenetic marks.

## Data Availability

No new data was created in this review. Data sharing not applicable.

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
