# Peer review of "Impact of 5-HT6 Receptor Subcellular Localization on Its Signaling and Its Pathophysiological Roles"

_cells, 2023, doi:10.3390/cells12030426_

Round 1
Reviewer 1 Report
The article of Chaumont-Dubel et al is about the expression of the 5-HT6 receptor during the development and their targeting to the cilia of neurons and glial cells. The authors recall the complex signalling pathways triggered by the stimulation (agonist dependent or via the constitutive activity) of 5-HT6 receptors, the triggered signalling pathways being dependent on the sub-cellular location of the receptor. They also discuss the motifs possibly involved in the specific cell targeting. Thus they gave a molecular and cellular picture of this singular receptor to stress out the points that it could have a fundamental role in the development of the CNS and could be involved in numerous pathologies.
The article is well written, well narrowed to a specific topic, and the illustrations are pretty and appropriate (although the photomicrographs are not so easy to analyse). The cited literature is appropriate. I honestly enjoyed reading the article. I have only some minor remarks.
Line 88: I’m not totally sure about the term “trafficked”
Line 93: has not
Line 151: the authors could mention the technique used by the authors of this article to catch “serotonin release” in the vicinity of cilia.
Line 173-174: references are not correctly reported
Line 175: pathophysiological
Line 177: mouse model of…
Line 183: would the authors mind to be more explicit as regards D72, D106 and so on?
Line 185: please adjust
Starting from line 186 to line 193: the authors mention some “antagonists” at 5-HT6R. I was wondering if these compounds could behave as inverse agonists.
Line 215: 5-HT6R
Line 254: , instead of .
Author Response
We thank the reviewer for the careful evaluation of our manuscript and constructive remarks.
We have taken into account all the points. All modifications appear in blue police in the manuscript.
The typos and grammar/language have been corrected.
Line 102 (previously 88): “trafficked” has been changed to "addressed"
Line 108 (previously 93): has not
Line 173 (previously 151): we added the requested precision
Line 195 ( previously 173-174): references are now correctly reported
Line 199 ( previously 175): pathophysiological is now correct
Line 201 (previously 177): mouse model of… has been added
Line 183: would the authors mind to be more explicit as regards D72, D106 and so on?
- We added an explanation for the role of the D106, F69, T70 and D72 mutants, as well as the reference of the papers first using these mutants (line 206-212)
Starting from line 186 to line 193: the authors mention some “antagonists” at 5-HT6R. I was wondering if these compounds could behave as inverse agonists.
- The reviewer is right, the compounds cited do act as inverse agonists toward the Gs pathway. This precision is now added in the text (line 215-217)
Line 236 (previously 215): 5-HT6R has been corrected
Line 287 (previously 254): punctuation is now correct
Reviewer 2 Report
This is a useful and informative review article that addresses the signaling functions of the 5-HT6 receptor based on its localization to the primary cilium or cell body. The authors summarize much of their own work but do provide balance in citing other relevant literature. This reviewer noticed a few instances in which the literature was not cited accurately, however. These are noted below. This is otherwise an interesting and timely piece.
Line 45- “…of the receptor in neuropathic pain [30]...” Reference #30 does not address neuropathic pain.
Line 84- The authors state, “Pluripotent embryonic stem cells do not possess a primary cilium, but they will form a primary cilium upon differentiation [58].” Reference #58 provides evidence that primary cilia are a general feature of undifferentiated human embryonic stem cell lines. Please clarify.
Line 207- “Overexpression of the 5-HT 6R in neurons can lead to ectopic receptor localization in the somato-dendritic compartment [40,46].” Reference #40 did not employ neuronal cells.
Line 258- Again regarding reference #40, the authors comments are a bit oversimplified. This paper showed that treatment with hedgehog pathway activators unmasked 5-HT-stimulated cAMP production by ciliary 5-HT6 receptors and ciliary dopamine receptors (D1R). The authors imply that the GPCR is inactive in the cilium with respect to AC coupling, but this is conditional.
Figure 1B- A more zoomed image (perhaps an inset?) that better shows what individual cilia look like would be helpful for the non-specialist reader.
Figure 3- The layout is initially a bit confusing. Figure 3A and 3B consist of two panes, giving the reader the sense that there are multiple states in the soma membrane in postnatal and adult neurons. If it were possible to merge the right and left panes (depicting one GPCR giving rise to multiple signaling pathways) the meaning might be more immediate.
Author Response
We thank the reviewer for the careful evaluation of our manuscript and constructive remarks.
We have taken into account all the points. All modifications appear in blue police in the manuscript, and you will find below a point by point reply to comments.
- Line 45 the reference was indeed not pointing toward the right study. This has been corrected.and we now reference Martin et al (ref 32)
- The reviewer is right, stem cells possess a primary cilium, and our statement was not correct. What we meant to convey is that they do not have a stable primary cilium, readily observable in culture. We have clarified this point, adding the reference for a study published in Cell by Phua et al linking cell cycle and primary cilium decapitation. corrections appear in line 94 to 99
- Line 237 (previously 207), the reviewer is correct, reference 40 uses mIMCD3 cells. We modified the sentence which now reads " Overexpression of the receptor in neurons or in mIMCD3 cell..."
- We apologize for the over-simplification of the conclusion of Jiang et al's study. We have modified the paragraph to describe the conditions in which the receptor can be coupled to Gs in the primary cilium (See line 287 to 295)
- We thank the reviewer for the suggestion to improve figure 1B. An inset has been added on the merge image to facilitate visualization of the cilium.
- For Figure 3A, our studies describe that the 5-HT6R interacts sequentially with Cdk5 or with GPRIN1. GPRIN1 is unable to interact with the Cdk5-phosphorylated receptor. Therefore, two types of receptor can exist at the plasma membrane, evolving from Cdk5 coupling to GPRIN1 coupling over time. We added an arrow between the two states to depict this sequential interaction over time. For figure 3B, we tried representing one single receptor linked to both pathways as suggested, but doing so, we felt we lost clarity. Furthermore, we do not know if both pathways can simultaneously interact with the same receptor and if receptors interacting with each pathways are in the same conformational state. We therefore propose to keep the layout presented but added these precisions in the figure legend. We hope that will be satisfactory.
Author Response
We thank the reviewer for careful evaluation of our manuscript and constructive suggestions. Modifications appear in blue in the revised manuscript
- We modified all the sentences which started with "This".
- Line 25 we removed the word chemical
- Line 39 To our knowledge, it is also the only serotonin receptor that has been identified in the primary cilium (Schou et al). A search in the CiliaCarta database, a compendium of all ciliary genes indicates retros only the 5-HT6 receptor when serotonin is used as a keyword. With references have been added to substantiate the claim. Furthermore, none of the CTS are conserved in the sequences of the other 5-HT receptors.
- Line 44 : the suggested change has been done
- We added the reference and a summary of the roles of the receptor in migration and morphogenesis. The new paragraph appears line 50-58
- Cartograph was replaced by mapping of as suggested (line 72)
- "Do not give information" was changed to "did not reveal" (line 73)
- The change was done (line 84)
- Ciliopathies are now defined, and examples removed (line 87-90)
- The reviewer is correct, the section was not completely accurate, and we apologize for that. We rewrote this part to take into account the suggestions of the reviewer. We added the possible role of the alanine, and the fact that the CTS are redundant. We also modified the figure which now indicate "partial targeting to the cilia" when only one CTS is mutated. (line 129-140 and line 147)
- See above.
- the terms relocates has been used to replace reintegrates. We added a sentence to clarify that at the embryonic stage, the receptor is in the cilium and he localization at the cell body is transient, only during the first few post-natal days (line 243-247)
- We now specify that the experiments were done in IMCD3 cells (line 235), and the reference for the Jiang et al study has been added.
- Soma has been replaced by cell body throughout the manuscript and in figure 3 as suggested
All additional points were corrected.